# GROUP DETR: FAST DETR TRAINING WITH GROUP-WISE ONE-TO-MANY ASSIGNMENT

## ABSTRACT

Detection Transformer (DETR) relies on one-to-one assignment for end-to-end object detection and lacks the capability of exploiting multiple positive object queries. We present a novel DETR training approach, named *Group DETR*, to support one-to-many assignment in a group-wise manner. To achieve it, we make simple modifications during training: (i) adopt $K$ groups of object queries; (ii) conduct decoder self-attention on each group of object queries with the same parameters; (iii) perform one-to-one assignment for each group, leading to $K$ positive object queries for each ground-truth object. In inference, we only use one group of object queries, making no modifications to model architectures and inference processes. Group DETR is a versatile training method and is applicable to various DETR variants. Our experiments show that Group DETR significantly speeds up the training convergences and improves the performances of various DETR-based methods.

## 1 INTRODUCTION

Detection Transformer (DETR) (Carion et al., 2020) achieves end-to-end detection without the need of non-maximum suppression (NMS) (Hosang et al., 2017). There are several designs: (i) adopt an encoder-decoder architecture based on transformer layers (Vaswani et al., 2017), (ii) introduce object queries, and (iii) perform one-to-one assignment[1] by conducting bipartite matching (Kuhn, 1955) between object predictions and ground-truth objects.

The original DETR suffers from the slow convergence issue and needs 500 training epochs to achieve good performance. Various solutions have been developed to accelerate the training from different aspects. For example, sparse transformers (Zhu et al., 2020b; Gao et al., 2021; Chen et al., 2022c; Roh et al., 2022) are adopted to replace dense transformers. Additional spatial modulations are introduced into object queries (Zhu et al., 2020b; Meng et al., 2021; Wang et al., 2022b; Yao et al., 2021; Liu et al., 2022a; Gao et al., 2022). Denoising modules are presented for stabilizing the object query and group-truth matching in the assignment process (Li et al., 2022; Zhang et al., 2022b).

In this paper, we propose a novel training approach *Group DETR* to accelerate DETR training convergence. Group DETR introduces *group-wise one-to-many assignment*. It assigns each ground-truth object to many positive object queries (one-to-many assignment[2]), and separate them into multiple independent groups, keeping only one positive object query per object (one-to-one assignment) in each group. To achieve it, we make simple modifications during training: (i) adopt $K$ groups of object queries; (ii) conduct decoder self-attention on each group of object queries with the same parameters; (iii) perform one-to-one assignment in each group, leading to $K$ positive object queries for each ground-truth object. The design achieves fast training convergence, maintaining the key DETR property: enabling end-to-end object detection without NMS. We only use one group of object queries in inference, and we do not modify either architectures or processes, bringing no extra cost compared with the original method.

Group DETR is a versatile training method and can be applied to various DETR-based models. Extensive experiments prove that our method is effective in achieving fast training convergence

---

[1]One-to-one assignment: one ground-truth object is only assigned to one object query.
[2]One-to-many assignment: each ground-truth object can be assigned to one or more positive object queries.

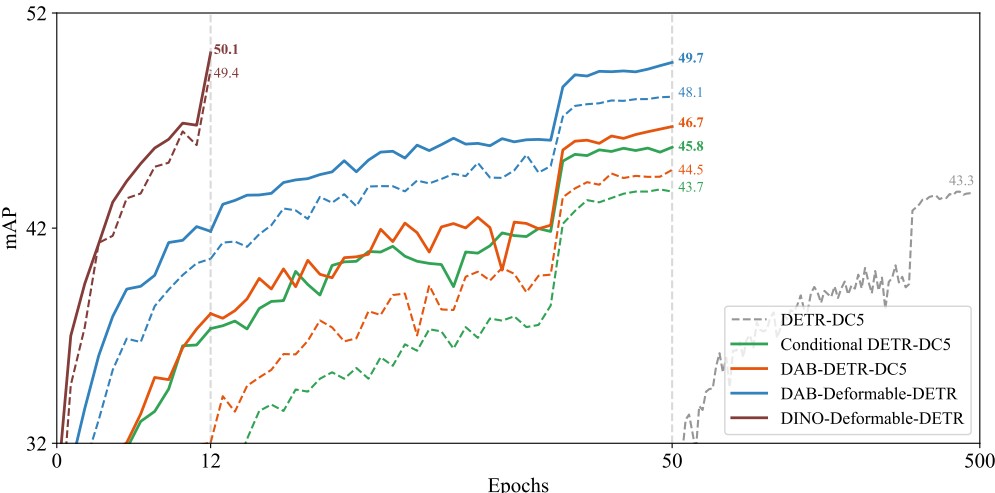

Figure 1: **Comparisons on training convergence curves.** We show the training convergence curves of various DETR-based models. All experiments are conducted on MS COCO (Lin et al., 2014) with ResNet-50 (He et al., 2016) as the backbone. More results and comparisons can be found in Table 1 and Table 2. Here, we use different colors to distinguish different models in the figure and apply dashed curves and bold curves to highlight the comparisons between baseline models and their Group DETR counterparts. Best view in color.

(convergence curves are shown in Figure 1). Group DETR obtains consistent improvements on various DETR-based methods (Meng et al., 2021; Liu et al., 2022a; Li et al., 2022; Zhang et al., 2022b). With a 12-epoch (1×) training schedule on MS COCO (Lin et al., 2014), Group DETR significantly improves Conditional DETR-C5 by **5.0** mAP. The non-trivial improvements hold when we adopt longer training schedules (*e.g.*, 36 epochs or 50 epochs). Moreover, Group DETR can easily outperform baseline models when applied to multi-view 3D object detection (Liu et al., 2022b;c) and instance segmentation (Cheng et al., 2021).

## 2 RELATED WORKS

**Acceleraing DETR training convergence.** The success of DETR (Carion et al., 2020) in object detection validates the potential to achieve elegant designs with transformers in computer vision. Since DETR (Carion et al., 2020) was proposed, its slow convergence issue has been a critical problem that many researchers (Bar et al., 2022; Wang et al., 2022a; Song et al., 2022; Roh et al., 2022) try to address.

Many works provide their solutions and achieve a 10× speed up for DETR. They mainly focus on proposing better transformer layers (Zhu et al., 2020b; Gao et al., 2021; Meng et al., 2021; Dai et al., 2021; Roh et al., 2022; Cao et al., 2022; Zhang et al., 2022a; Chen et al., 2022d) and designing new types of object queries (Zhu et al., 2020b; Meng et al., 2021; Wang et al., 2022b; Yao et al., 2021; Liu et al., 2022a; Gao et al., 2022). DN-DETR (Li et al., 2022) and DINO (Zhang et al., 2022b) attribute the slow convergence issue to the instability of bipartite matching (Kuhn, 1955).They present auxiliary query denoising tasks to speed up the DETR training convergence. Unlike previous approaches, we show that assignment methods are critical to fast DETR training convergence. We propose Group DETR to support one-to-many assignment in a group-wise manner, which can be achieved by simple modifications during training compared with DETR.

**One-to-many assignment and one-to-one assignment.** One-to-many assignment is widely adopted in modern detectors (Redmon et al., 2016; Ren et al., 2015; Liu et al., 2016; He et al., 2017; Lin et al., 2017; Cai & Vasconcelos, 2018; Chen et al., 2019; Tian et al., 2019; Zhang et al., 2020; Zhu et al., 2020a; Kim & Lee, 2020; Bochkovskiy et al., 2020; Chen et al., 2021; Ge et al., 2021). It produces duplicate predictions and needs NMS (Hosang et al., 2017; Bodla et al., 2017) for post-processing. DETR (Carion et al., 2020) explores an alternative way (one-to-one assign-

ment) and achieves end-to-end detection, removing the need for NMS. Recent studies (Wang et al., 2021; Sun et al., 2021b;a) show that one-to-one assignment is a key factor in achieving end-to-end detection. Differently, we find that the one-to-one assignment impacts the training convergence of DETR-based methods (Carion et al., 2020) and we focus on exploiting assignment methods to speed up DETR training in this paper.

## 3 GROUP DETR

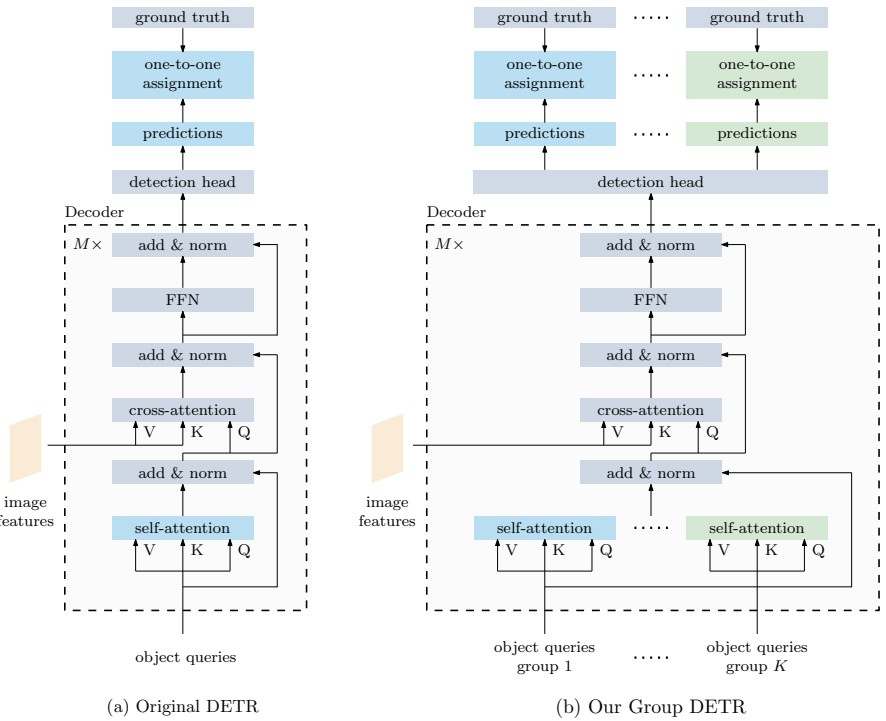

(a) Original DETR                    (b) Our Group DETR

Figure 2: **Decoder architectures of DETR and Group DETR for training.** The key differences from DETR include: Group DETR feeds $K$ groups of queries to the decoder, conducts self-attention on each group of object queries with the shared parameters, and makes one-to-one assignment for each group.

Group DETR is a training approach for accelerating the DETR training convergence. We make simple modifications during training and adopt the same architecture and inference process for inference. Figure 2 illustrates the decoder parts of DETR (Carion et al., 2020) and our Group DETR for training.

### 3.1 DETR

DETR has three key designs: (i) adopt a transformer encoder-decoder architecture (Vaswani et al., 2017), (ii) introduce object queries, and (iii) perform one-to-one assignment by conducting bipartite matching between object predictions and ground-truth objects.

**DETR architecture.** The DETR architecture consists of a backbone (*e.g.*, ResNet (He et al., 2016), Swin Transformer (Liu et al., 2021), or others (Dosovitskiy et al., 2021; Liu et al., 2022d; Chen et al., 2022a; He et al., 2022; Chen et al., 2022b)), a transformer encoder, a transformer decoder, and object class and box position predictors (Carion et al., 2020). Figure 2 (a) shows the architecture of the transformer decoder in DETR. The image features are extracted by the backbone and the transformer encoder layers. The transformer decoder takes $N$ object queries $\{\mathbf{q}_1, \ldots, \mathbf{q}_N\}$ as input. It performs self-attention on object queries, aggregates the image features to refine the

query embeddings by conducting cross-attention, and adds an FFN to get the output query embeddings. The output query embeddings are fed into detection heads to produce $N$ object predictions.

**One-to-one assignment.** In model training, DETR perform one-to-one assignment to find the learning target for each object prediction. It employs the Hungarian algorithm (Kuhn, 1955) to find an optimal bipartite matching $\hat{\sigma}$ between the predictions and the ground-truth objects:

$$\hat{\sigma} = \arg\min_{\sigma \in \xi_N} \sum_{i=1}^{N} \mathcal{C}_{\text{match}}(y_i, \hat{y}_{\sigma(i)}), \tag{1}$$

where $\xi_N$ is the set of permutations of $N$ elements and $\mathcal{C}_{\text{match}}(y_i, \hat{y}_{\sigma(i)})$ is the matching cost (Carion et al., 2020) between the ground truth $y_i$ and the prediction with index $\sigma(i)$.

### 3.2 GROUP DETR

Group DETR makes simple modifications during training compared with DETR (Figure 2 (b)): (i) adopt $K$ groups of object queries; (ii) conduct decoder self-attention on each group of object queries with the same parameters; (iii) perform one-to-one assignment in each group (*group-wise one-to-many assignment*), leading to $K$ positive object queries for each ground-truth object.

**$K$ groups of object queries.** There are $K$ groups of queries in the proposed Group DETR:

$$\mathbb{G}^1 = \{\mathbf{q}_1^1, \dots, \mathbf{q}_N^1\}, \tag{2}$$

$$\dots\dots$$

$$\mathbb{G}^K = \{\mathbf{q}_1^K, \dots, \mathbf{q}_N^K\}. \tag{3}$$

The total $K \times N$ object queries are concatenated and fed to transformer decoder layers (as shown in Figure 2 (b)).

**Group-wise decoder self-attention.** We perform group-wise self-attention in the transformer decoder layers. This leads to that the object queries do not interact with the queries across groups. The pseudocode of group-wise self-attention is in Algorithm 1.

**Group-wise one-to-many assignment.** We apply one-to-one assignment to each group independently and we could get $K$ matching results:

---

**Algorithm 1** Pseudocode of one Group DETR decoder layer in python-style

```
# self_atten: self-attention in decoder
# cross_atten: cross-attention in decoder
# ffn: FFN in decoder
# X: output image features of the encoder
# Q: object queries, with size (KxN, B, C)
# N, K, B, C: object query number, group number,
    batch size, feature dimension

# Group DETR
if training:
    # split object queries to K groups
    Q_list = Q.split(N, dim=0) # a list of K tensors
    group_Q = cat(Q_list, dim=1) # (N, KxB, C)

    # group-wise self-attention
    out = self_atten(group_Q) # (N, KxB, C)
    # concat all groups: (KxN, B, C)
    out = cat(out.split(B, dim=1), dim=0)

    # cross-attention and ffn
    out = ffn(cross_atten(out, X))
else:
    # in inference, only one group is kept
    Q = Q[:N] # (N, B, C)

    # self-attention, cross-attention, and ffn
    out = self_atten(Q)
    out = ffn(cross_atten(out, X))
```

---

$$\hat{\sigma}^1 = \arg\min_{\sigma \in \xi_N} \sum_{i=1}^{N} \mathcal{C}_{\text{match}}(y_i, \hat{y}_{\sigma(i)}^1), \tag{4}$$

$$\dots\dots$$

$$\hat{\sigma}^K = \arg\min_{\sigma \in \xi_N} \sum_{i=1}^{N} \mathcal{C}_{\text{match}}(y_i, \hat{y}_{\sigma(i)}^K), \tag{5}$$

where $\hat{\sigma}^K$ and $\hat{y}_{\sigma(i)}^K$ are the optimal matching result and the prediction of the $K$-th group, respectively. During training, each group will calculate the loss (Carion et al., 2020) independently in Group DETR. The final training loss is the average of $K$ groups.

**Model inference.** We adopt the same architectures and processes as the baseline models in inference. According to our experiments, every group can achieve similar results in Group DETR (Table 5). We simply use the first group of object queries.

| Model | w/ Group DETR | mAP | $AP_s$ | $AP_m$ | $AP_l$ |
|---|---|---|---|---|---|
| Conditional DETR-C5 | | 32.6 | 14.7 | 35.0 | 48.3 |
| Conditional DETR-C5 | ✓ | 37.6 (+5.0) | 18.2 | 40.7 | 55.9 |
| Conditional DETR-DC5 | | 36.4 | 18.0 | 39.6 | 52.5 |
| Conditional DETR-DC5 | ✓ | 41.2 (+4.8) | 21.4 | 45.0 | 58.7 |
| DAB-DETR-C5 | | 35.2 | 16.7 | 38.6 | 51.6 |
| DAB-DETR-C5 | ✓ | 39.1 (+3.9) | 19.7 | 42.5 | 56.8 |
| DAB-DETR-DC5 | | 37.5 | 19.4 | 40.6 | 53.2 |
| DAB-DETR-DC5 | ✓ | 41.9 (+4.4) | 23.3 | 45.6 | 58.4 |
| DN-DETR-C5 | | 38.6 | 17.9 | 41.6 | 57.7 |
| DN-DETR-C5 | ✓ | 40.6 (+2.0) | 19.8 | 43.9 | 59.4 |
| DN-DETR-DC5 | | 41.9 | 22.2 | 45.1 | 59.8 |
| DN-DETR-DC5 | ✓ | 44.5 (+2.6) | 25.9 | 48.2 | 62.2 |
| DAB-Deformable-DETR | | 44.2 | 27.5 | 47.1 | 58.6 |
| DAB-Deformable-DETR | ✓ | 45.7 (+1.5) | 28.1 | 49.0 | 60.6 |
| DINO-Deformable-DETR | | 49.4 | 32.3 | 52.5 | 63.2 |
| DINO-Deformable-DETR | ✓ | **50.1** (+0.7) | 32.4 | 53.2 | 64.7 |

Table 1: **Results with a** 12-**epoch training schedule on MS COCO.** All experiments adopt ResNet-50 (He et al., 2016) as the backbone. We highlight the improvements brought by Group DETR on various DETR-based methods. Note that we do not use multiple patterns (Wang et al., 2022b) in our experiments. For DN-DETR in the table, we use the improved version of DN (dynamic DN groups (Zhang et al., 2022b)) and set the DN number to 100 (more results about the DN number can be found in Appendix B Table 8). Thus, the baseline results of DN-DETR are slightly different from the ones (with 3 patterns) reported in the original paper (Li et al., 2022). For DINO-Deformable-DETR, we adopt the *4scale* version (Zhang et al., 2022b).

## 4 EXPERIMENTS

We demonstrate the effectiveness of our Group DETR on object detection, instance segmentation and multi-view 3D object detection. We adopt the training settings and hyper-parameters same as the baseline models, including learning rate, optimizer, pre-trained model, initialization methods, and data augmentations[3]. The number of queries in each group is the same as the baselines.

### 4.1 OBJECT DETECTION

We verify our approach over object detection on MS COCO (Lin et al., 2014) and make comparison with representative DETR-based methods, including Conditional DETR (Meng et al., 2021), DAB-DETR (Liu et al., 2022a), DN-DETR (Liu et al., 2022a; Zhu et al., 2020b), and DINO-Deformable-DETR (Zhang et al., 2022b; Zhu et al., 2020b). We report the results with 12-epoch (1×) and 50-epoch training schedules, as well as the comparison in terms of convergence curve.

**Results with a standard** 1× **schedule.** Table 1 report the results. Group DETR gives consistent improvements over all baseline models. In comparison to query design methods, Group DETR significantly boosts detection performance when applied to Conditional DETR (+5.0 mAP for C5 and +4.8 mAP for DC5) (Meng et al., 2021), DAB-DETR (+3.9 mAP for C5 and +4.4 mAP for DC5) (Liu et al., 2022a), and DAB-Deformable-DETR (+1.5 mAP for multiple levels of feature maps) (Liu et al., 2022a; Zhu et al., 2020b).

In comparison to the matching stabilization methods, Group DETR can also give non-trivial gains over DN-DETR (+2.0 mAP for C5 and +2.6 mAP for DC5) (Li et al., 2022) and DINO-Deformable-DETR (+0.7 mAP for multiple levels of feature maps) (Zhang et al., 2022b; Zhu et al.,

---

[3]We may adjust the batch size according to the GPU memory. Note that we will retrain the baseline model with the same batch size to make fair comparisons when conducting experiments.

| Model | w/ Group DETR | mAP | $AP_s$ | $AP_m$ | $AP_l$ |
|---|---|---|---|---|---|
| Conditional DETR-C5 | | 40.9 | 20.5 | 44.2 | 59.6 |
| Conditional DETR-C5 | ✓ | 43.4 (+2.5) | 23.0 | 47.3 | 62.3 |
| Conditional DETR-DC5 | | 43.7 | 23.9 | 47.6 | 60.1 |
| Conditional DETR-DC5 | ✓ | 45.8 (+2.1) | 26.8 | 49.7 | 63.1 |
| DAB-DETR-C5 | | 42.2 | 21.5 | 45.7 | 60.3 |
| DAB-DETR-C5 | ✓ | 44.5 (+2.3) | 24.2 | 48.5 | 63.2 |
| DAB-DETR-DC5 | | 44.5 | 25.3 | 48.2 | 62.3 |
| DAB-DETR-DC5 | ✓ | 46.7 (+2.2) | 27.6 | 50.9 | 64.0 |
| DN-DETR-C5 | | 44.0 | 23.9 | 47.7 | 62.9 |
| DN-DETR-C5 | ✓ | 45.4 (+1.4) | 25.1 | 49.3 | 63.8 |
| DN-DETR-DC5 | | 47.5 | 27.9 | 50.7 | 65.9 |
| DN-DETR-DC5 | ✓ | 48.0 (+0.5) | 29.3 | 52.1 | 65.4 |
| DAB-Deformable-DETR | | 48.1 | 31.4 | 51.4 | 63.4 |
| DAB-Deformable-DETR | ✓ | 49.7 (+1.6) | 31.4 | 52.5 | 65.6 |
| DINO-Deformable-DETR | | 50.9 | 34.6 | 54.1 | 64.6 |
| DINO-Deformable-DETR | ✓ | 51.3 (+0.4) | 34.7 | 54.5 | 65.3 |
| DINO-Deformable-DETR-Swin-L | ✓ | **58.4** | 41.0 | 62.5 | 73.9 |

Table 2: **Results with a** $50$**-epoch training schedule on MS COCO.** We adopt the training schedule of 50 epochs in the table, while for DINO-Deformable-DETR (Zhang et al., 2022b; Zhu et al., 2020b), we train 36 epochs by following the settings in the original paper (Zhang et al., 2022b). Except that we use Swin-Large (Liu et al., 2021) in the last row of the table, we use ResNet-50 (He et al., 2016) as the backbone. As in Table 1, we do not use multiple patterns and apply dynamic DN groups in DN-DETR. For DINO-Deformable-DETR, we adopt the *4scale* version (Zhang et al., 2022b).

2020b), even though they already achieve strong results by introducing auxiliary query denoising tasks in model training.

**Results with a** $50$**-epoch training schedule.** Compared with the original DETR model, the above DETR-based methods achieve a $10\times$ speed up on training. They provide good results with a 50 epochs training schedule. We show the effectiveness of Group DETR under this setting. As presented in Table 2, Group DETR can also outperform baseline models by large margins. When we adopt a stronger backbone, Swin-Large (Liu et al., 2021), we can achieve 58.4 mAP (0.4 mAP higher than its baseline DINO-Deformable-DETR (58.0 mAP with Swin-Large)), which verifies the generalization ability of our Group DETR.

**Convergence curves.** We report the convergence curves in Figure 1. We give each method two convergence curves of the baseline model and the baseline with Group DETR using dashed curves and bold curves. The comparisons in Figure 1 support that Group DETR gives a further speed up on DETR training convergence.

## 4.2 MULTI-VIEW 3D OBJECT DETECTION AND INSTANCE SEGMENTATION

**Multi-view 3D object detection.** We adopt PETR (Liu et al., 2022b) and PETR v2 (Liu et al., 2022c) as our baseline models. Table 3 shows that Group DETR brings significant gains to PETR and PETR v2. When we train PETR v2 with a longer training schedule (36 epochs), we obtain more improvements on both nuScenes Detection Score (NDS) and mAP on the nuScenes `val` set (Caesar et al., 2020).

**Instance segmentation.** We adopt Mask2Former (Cheng et al., 2021) as the baseline and apply Group DETR to it. In Table 4, we provide comparisons of different training schedules. Group DETR achieves non-trivial improvements on Mask2Former.

| Model | w/ Group DETR | Schedule | NDS | mAP |
|---|---|---|---|---|
| PETR | | 24e | 42.0 | 37.4 |
| PETR | ✓ | 24e | 45.0 (+3.0) | 38.8 (+1.4) |
| PETR v2 | | 24e | 50.3 | 40.7 |
| PETR v2 | ✓ | 24e | 51.3 (+1.0) | 41.9 (+1.2) |
| PETR v2 | | 36e | 50.8 | 41.3 |
| PETR v2 | ✓ | 36e | **52.3** (+1.5) | **42.7** (+1.4) |

Table 3: **Results on multi-view 3D object detection.** All experiments are conducted on the nuScenes `val` set (Caesar et al., 2020). We train these experiments with VoVNetV2 (Lee & Park, 2020) as the backbone and with the image size of $800 \times 320$. We follow all the settings and hyperparameters of PETR and PETR v2.

| Group DETR | Schedule | $mAP^m$ | $AP^m_s$ | $AP^m_m$ | $AP^m_l$ |
|---|---|---|---|---|---|
| | 12e | 38.5 | 17.6 | 41.4 | 60.4 |
| ✓ | 12e | 39.7 (+1.2) | 18.7 | 42.8 | 60.8 |
| | 50e | 43.7 | 23.4 | 47.2 | 64.8 |
| ✓ | 50e | **44.0** (+0.3) | 23.8 | 47.1 | 65.1 |

Table 4: **Results on instance segmentation.** We adopt Mask2Former (Cheng et al., 2021) as the baseline and report the mask mAP ($mAP^m$) on the MS COCO `val` split for instance segmentation.

## 4.3 ABLATION STUDIES

We conduct ablation studies on object detection by using Conditional DETR-C5 (Meng et al., 2021) as our baseline model. The studies include: the influence of group number, the performance in each group, the assignment scheme, and the group design.

**Influence of group number.** Figure 3 shows the influence of the number of groups $K$ in Group DETR. The detection performance improves when increasing the number of groups, and saturates when the group number ($K$) is greater than 11. Thus, we adopt $K = 11$ in Group DETR in our experiments.

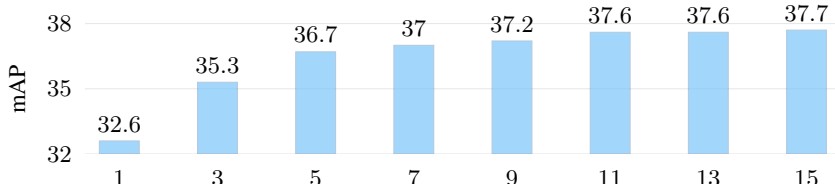

Figure 3: **Influence of group number ($K$) in Group DETR.** As the number of groups increases, continuous improvement could be achieved compared to the baseline model.

**Performance in each group.** Table 5 gives the performance of all groups in Group DETR. Each group can achieve similar results, which is consistent with the design of independent groups. In other experiments, we simply report the result of the first group.

**Assignment.** Figure 4 and Table 6 study the training convergence and performance by keeping the same number (3300) of total object queries about three cases: single-group one-to-one assignment ($K = 1$), single-group one-to-many assignment ($K = 1$ with 11 positive object queries for each ground-truth object), and group-wise one-to-many assignment with $K = 11$ groups (Group DETR). Figure 4 shows that Group DETR and single-group one-to-many assignment give significantly faster convergence speeds than single-group one-to-one assignment.

Different from single-group one-to-one assignment in DETR (Carion et al., 2020), single-group one-to-many assignment highly depends on the post-processing step NMS (Table 6). Our Group DETR

| group-1 | group-2 | group-3 | group-4 | group-5 | group-6 | group-7 | group-8 | group-9 | group-10 | group-11 |
|---------|---------|---------|---------|---------|---------|---------|---------|---------|----------|----------|
| **37.6** | 37.5 | 37.4 | 37.5 | 37.6 | 37.5 | 37.6 | 37.4 | 37.5 | 37.6 | 37.5 |

Table 5: **Performance in each group.** We show the mAP on the MS COCO *val* split of different groups in Group DETR. The results obtained by different groups are similar.

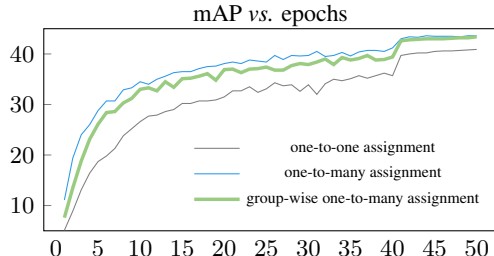

| Assignment | NMS | mAP |
|------------|-----|-----|
| one-to-one | **w/o** | **41.5** |
| one-to-one | w/ | 41.4 |
| one-to-many | w/o | 7.2 |
| one-to-many | **w/** | **43.6** |
| group-wise one-to-many | **w/o** | **43.4** |
| group-wise one-to-many | w/ | 43.3 |

Figure 4: **Training convergence curves of different assignment methods.** Multiple positive object queries converge faster than one positive object query per object.

Table 6: **Comparisons of different assignment methods with and without NMS.** Results are obtained with a 50-epoch training schedule. We set a threshold of 0.7 in NMS following DETR.

w/o NMS achieves almost the same performance as single-group one-to-many assignment w/ NMS, and the inference of our Group DETR is more efficient than it. It is as expected that our Group DETR performs better than single-group one-to-one assignment.

**Group design.** DN-DETR and our Group DETR both adopt the group design and focus on different aspects: stabilizing the prediction and group-truth matching (DN-DETR), and exploiting multiple positive predictions for one ground-truth object (ours), respectively. The performance comparison for DN-DETR and Group DETR in Table 7 shows that Group DETR is superior than DN-DETR. The result of the combination of DN-DETR and Group DETR further improves the performance to 40.6 mAP, implying that the two approaches are complementary.

| Group Design | None | DN-DETR* | Group DETR | DN-DETR + Group DETR |
|--------------|------|----------|------------|---------------------|
| mAP | 35.2 | 38.8 | 39.1 | **40.6** |

Table 7: **Comparisons on group designs.** We adopt the DAB-DETR (Liu et al., 2022a) as the baseline model (the one with 'None' group design in the table) and train all models with a 12-epoch schedule on MS COCO. * represents that we report the best results of DN-DETR among different numbers of denoising queries (more results are provided in Appendix B Table 8).

## 4.4 ANALYSES

**The object queries are distributed similarly for all groups.** We study the distributions of object queries in each group with conditional DETR as an example. Figure 5 depicts 2D reference points (positions) corresponding to the object queries, with one color for one group. We can see that the reference point distributes similarly for all the groups. This provides an explanation to that each group of object queries gives similar detection performances in Table 5.

**Visualization of positive object queries.** We visualize the positions of positive object queries in all groups in Figure 6. The visualization

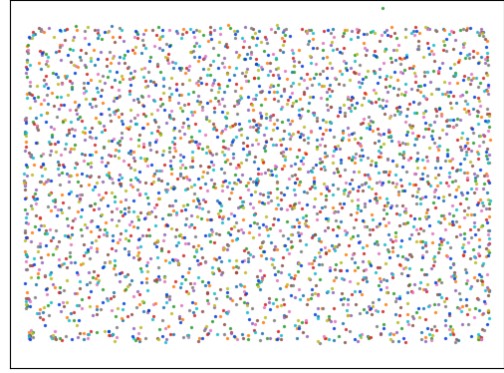

Figure 5: **Distribution of all groups of object queries with different colors.**

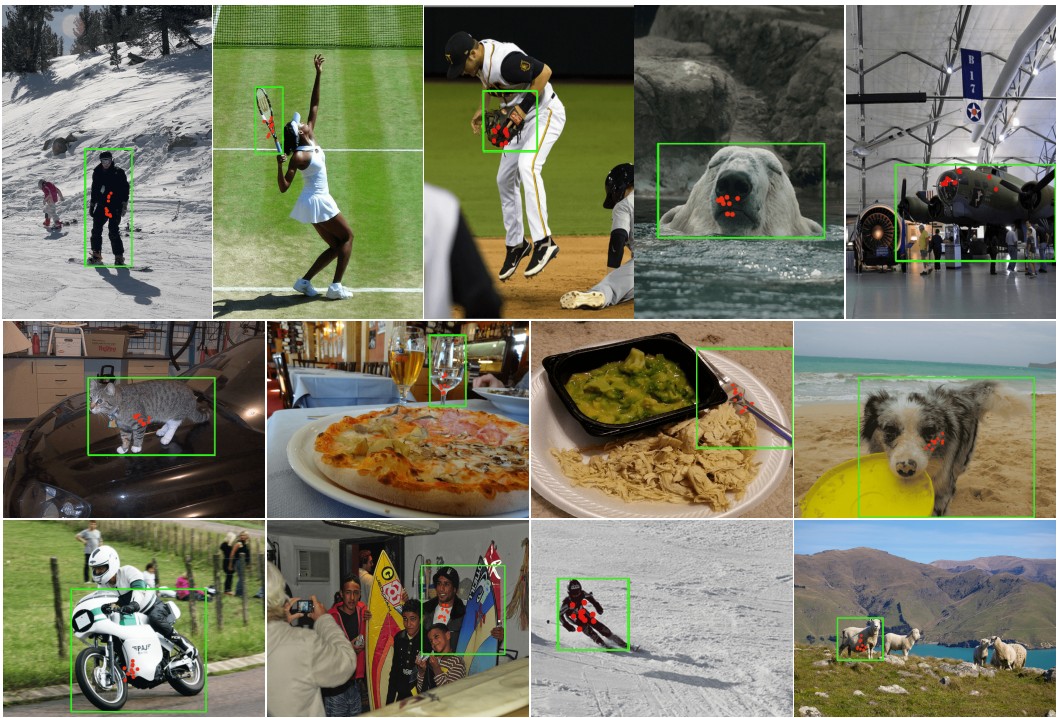

Figure 6: **Visualization of positive object queries.** To give a neat visualization, we only show one object per image. Each ground-truth box (green bounding box) is assigned to multiple positive queries (red points). We train Group DETR based on Conditional-DETR-C5 for 50 epochs. The number of groups is 11 here. In the figure, the positive queries (red points) may overlap. Best view in color and zoom in.

shows that the positions of positive object queries are distributed in a certain region on *the object instance*. This region is learned by the model and is somehow different from manually selecting a center region within the bounding box. This is reasonable, since the center of the bounding box may not have enough information about the object instance. According to the visualization, the positions of all positive object queries are considered good ones by the model to predict the ground-truth objects. In the one-to-one assignment, only one of these object queries can be set as positive for the object, while others are negatives. The model needs to distinguish the differences among these object queries during training, which impact model learning, leading to slow training convergence. With Group DETR, all these object queries are set as positives, which gives stronger supervision signals in training, thereby improving training efficiency and speeding up training convergence.

**Training memory and training time.** Compared with the baseline models, Group DETR adopts more object queries in the transformer decoder during training. It is expected that Group DETR will increase training memory and training time. Thanks to the parallel computation with $K$ groups (shown in Algorithm 1), we only observe a $\sim15\%$ increase in training time. For instance, Group DETR increases the time for training one epoch from 47 minutes (96 minutes) to 51 minutes (108 minutes) on Conditional DETR-C5 (-DC5), from 61 minutes (121 minutes) to 70 minutes (135 minutes) on DN-DETR-C5 (-DC5). Group DETR augments the training memory of Conditional DETR-C5 (-DC5) from 7.1 G (13.7 G) to 16.8 G (23.0 G), the training memory of DN-DETR-C5 (-DC5) from 6.8 G (10.1 G) to 19.8 G (20.4 G). All the above training time and training memory are measured with 8 Tesla A100 GPUs.

## 5 CONCLUSION

In this paper, we present a simple yet effective approach, Group DETR, to accelerate DETR training convergence. We study different assignment methods and propose a novel group-wise one-to-many assignment. It has shown positive results in a variety of DETR-based methods and vision tasks.

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

# APPENDIX

## A    DATASETS AND EVALUATION METRICS

We perform the object detection and instance segmentation experiments on the COCO 2017 (Lin et al., 2014) dataset, which contains about 118K training (`train`) images and 5K validation (`val`) images. Following the common practice, we report the standard mean average precision (mAP) result (box mAP for object detection and mask mAP for instance segmentation) on the COCO validation dataset under different IoU thresholds and object scales.

We perform multi-view 3D object detection experiments on the nuScenes (Caesar et al., 2020) dataset, which contains 1000 driving sequences. There are 700 for `train` set, 150 for `val` set and 150 for `test` set. We report the standard nuScenes Detection Score (NDS) and mean Average Precision (mAP) result on the nuScenes `val` set.

## B    ADDITIONAL RESULTS

**Results of DN-DETR with different number of denoising queries.**    We conduct experiments with different numbers of denoising queries in DN-DETR (Li et al., 2022). The results in Table 8 suggest that increasing the number of denoising queries can not achieve further improvements and show unstable performances. The effects of denoising queries differ from the ones of Group DETR (Figure 3). The denoising queries mainly aim to solve the instability in the matching process, while our Group DETR aims to exploit multiple positive queries for one ground-truth object. We choose to use 100 denoising queries in our experiments in Table 1 and Table 2 by following the setting in the original paper (Zhang et al., 2022b).

| #Denoising Queries | 100 | 300 | 600 | 900 | 1200 | 1500 | 1800 | 2100 | 2400 | 2700 | 3000 | 3300 |
|---|---|---|---|---|---|---|---|---|---|---|---|---|
| mAP | 38.6 | 38.8 | 37.8 | 38.7 | 38.5 | 38.1 | 38.7 | 37.9 | 38.1 | 38.7 | 38.1 | 38.7 |

Table 8: **Results of DN-DETR with different number of denoising queries.** We show the detection performances (mAP) on MS COCO (Lin et al., 2014) of adopting different number of denoising queries in DN-DETR.

