# OpenReview forum: "Group DETR: Fast DETR Training with Group-Wise One-to-Many Assignment"
_ICLR.cc/2023/Conference — Submitted to ICLR 2023_

### Official Review · Reviewer_ySfN · 2022-10-21

**Confidence:** 5
**Correctness:** 4
**Technical Novelty And Significance:** 2
**Empirical Novelty And Significance:** 2
**Recommendation:** 5

**Clarity, Quality, Novelty And Reproducibility:**

As mentioned in the strengths section, this paper is well-written. However, no provide codes can be found, I only can see the pdf file, so I am not sure about the reproducibility. The paper proposes a novel but has similar effectiveness to other prior improvements of DETR, not a significant one.

**Strength And Weaknesses:**

*Strengths*
+ The paper is clearly written and easy to follow.
+ The proposed method is simple.
+ Extensive experiments have been conducted to verify the performance of the proposed method.

*Weaknesses*
+ This is not mandatory but it would be better to cite the Hybrid DETR (H-DETR) paper: https://arxiv.org/abs/2207.13080 and have a discussion to show to clear difference between the two.
+ The proposed method is orthogonal to the previous improvements of DETR so that when adding group-wise one-to-many assignments to others the performance is improved with a cost in training memory and time. There is no clear significant improvement over prior improvement. For example in Tab. 1, comparing the DN-DETR on top of the DAB-DETR with the denoising part and the Group-DETR on top of the DAB-DETR, the DN-DETR and Group-DETR are comparable (+3.4 vs +3.9 for C5 backbone and +4.4 vs +4.4 for DC5 backbone). Also, when adding Group-DETR on top of DINO, the improvement becomes less significant. In other words, adding Group-DETR on top of similar improvements results in less effectiveness.
+ The experiments are quite not fair. As Group-DETR increases the number of queries K times (K is the number of groups), it takes a longer time to train an epoch. Therefore, it would be better to compare the results of the proposed method with the previous methods on the same training hours, not the same number of epochs.
+ It is not clear why adding new groups improve the performance since there are more negative queries forcing in-object queries to be negative as well. The authors should have a better explanation.
+ Failure cases and limitations of the proposed method should be added since the proposed approach is not perfect.


**Summary Of The Paper:**

This paper proposes a new training strategy to improve the performance of DETR-based object detectors. The authors indicate the main reason for the low convergence rate of DETR-based object detectors is in the one-to-one matching and propose a simple strategy of group-wise one-to-many assignment to address. It is a simple plug-and-play strategy that can be applied to any DETR-based object detectors in the training phase. Extensive experiments have been conducted on 2D object detection (MS COCO), instance segmentation (MS COCO), and multi-view 3D object detection (Nuscenes) to show the advantage of the proposed approach.

**Summary Of The Review:**

+ The proposed method is simple and achieved impressive results on several DETR-based detectors and multiple datasets. However, there is a trade-off between performance versus runtime and memory consumption.
+ The idea of group-wise one-to-many assignments is quite similar to H-DETR. I am happy to increase this paper’s score if the authors provide experiments with the suggested setting above.

---

> ### Author Response · Authors · 2022-11-17
> **Response to reviewer ySfN (1/2)**
>
> Thanks a lot for your time and feedback. We give the responses to the concerns below.
>
> ___
>
> **Q1: Cite the Hybrid DETR (H-DETR) paper and have a discussion to show a clear difference between the two.**
>
> **A1:** Thanks for the suggestion. H-DETR is a concurrent work with our Group DETR. Group DETR and H-DETR both try to speed up DETR training convergence by proposing novel assignment methods. The difference between the two lies in the proposed assignment methods. Group DETR introduce group-wise one-to-many assignment and share the same modeling in each independent group. While H-DETR proposes hybrid matching, which consists of two groups, and these two groups are not the same as each other. One group uses one-to-one assignment and another uses one-to-many assignment with more object queries.
>
> We will add this discussion to Related Works.
>
> ___
>
> **Q2: Comparing the DN-DETR on top of the DAB-DETR with the denoising part and the Group-DETR on top of the DAB-DETR, the DN-DETR and Group-DETR are comparable (+3.4 vs +3.9 for C5 backbone and +4.4 vs +4.4 for DC5 backbone). Also, when adding Group-DETR on top of DINO, the improvement becomes less significant. In other words, adding Group-DETR on top of similar improvements results in less effectiveness.**
>
> **A2:** Our Group DETR and DN-DETR focus on different aspects (Group DETR aims to explore assignment methods, while DN-DETR focuses on stabilizing the prediction and ground-truth matching). We admit that, when using DAB-DETR as the baseline, both Group DETR and DN-DETR achieve similar improvements. But, Group DETR can also bring further gains on DN-DETR (+2.0/+1.4 mAP on DN-DETR-C5 with 12/50 epochs of training), implying that these two approaches can be complementary.
>
> In addition, DINO is a highly-optimized DETR variant. It adopts a two-stage process with mixed query selection, adds contrastive denoising queries (CDN), and proposes a look-forward-twice (LFT) scheme based on DAB-Deformable-DETR. It already speeds up the DETR training convergence and achieves 49.4 mAP with ResNet-50 as the backbone in 12 epochs. It is nontrivial to achieve gains on it with respect to speeding up training convergence. Group DETR brings a 0.7 mAP gain to DINO with 12 epochs and a 0.4 mAP gain with 36 epochs. These gains are non-negligible. Moreover, when using a strong backbone like Swin-Large, Group DETR also brings a 0.4 mAP gain to DINO. The results and analysis show the effectiveness of Group DETR.
>
> ___
>
> **Q3: It would be better to compare the results of the proposed method with the previous methods on the same training hours, not the same number of epochs.**
>
> **A3:** Thanks for the constructive suggestion. We conduct experiments on Conditional DETR, DAB-DETR, and DN-DETR. We train these baselines for longer training times, 15 epochs and 60 epochs, which give 25\% and 20\% more training epochs than 12 epochs and 50 epochs, respectively. The results are shown in the tables below.
>
> | Method  | w/ Group DETR | epochs  | mAP  | AP$_s$ | AP$_m$ | AP$_l$  |
> | ------------ | ------------ | ------------ | ------------ | ------------ | ------------ | ------------ |
> | Conditional DETR-C5  |  $\times$ | 12e | 32.6  | 14.7  | 35.0  | 48.3  |
> | Conditional DETR-C5  |  $\times$ | **15e** | 34.4  | 15.1  | 37.3  | 51.3  |
> | Conditional DETR-C5  |  √ | 12e | **37.6**  | 18.2  | 40.7  | 55.9  |
> ||
> | DAB-DETR-C5  |  $\times$ | 12e | 35.2  | 16.7  | 38.6  | 51.6  |
> | DAB-DETR-C5  |  $\times$ | **15e** | 36.3  | 17.1  | 39.4  | 52.5  |
> | DAB-DETR-C5  |  √ | 12e | **39.1**  | 19.7  | 42.5  | 56.8  |
> ||
> | DN-DETR-C5  |  $\times$ | 12e | 38.6  | 17.9  | 41.6  | 57.7  |
> | DN-DETR-C5  |  $\times$ | **15e** | 39.3  | 18.2  | 42.5  | 58.2  |
> | DN-DETR-C5  |  √ | 12e | **40.6**  | 19.8  | 43.9  | 59.4  |
>
>
> | Method  | w/ Group DETR | epochs  | mAP  | AP$_s$ | AP$_m$ | AP$_l$  |
> | ------------ | ------------ | ------------ | ------------ | ------------ | ------------ | ------------ |
> | Conditional DETR-C5  |  $\times$ | 50e | 40.9  | 20.5  | 44.2  | 59.6  |
> | Conditional DETR-C5  |  $\times$ | **60e** | 41.6  | 21.4  | 45.1  | 60.0  |
> | Conditional DETR-C5  |  √ | 50e | **43.4**  | 23.0  | 47.3  | 62.3  |
> ||
> | DAB-DETR-C5  |  $\times$ | 50e | 42.2  | 21.5  | 45.7  | 60.3  |
> | DAB-DETR-C5  |  $\times$ | **60e** | 42.9  | 22.8  | 46.4  | 61.9  |
> | DAB-DETR-C5  |  √ | 50e | **44.5**  | 24.2  | 48.5  | 63.2  |
> ||
> | DN-DETR-C5  |  $\times$ | 50e | 44.0  | 23.9  | 47.7  | 62.9  |
> | DN-DETR-C5  |  $\times$ | **60e** | 44.6  | 24.3  | 48.6  | 63.5  |
> | DN-DETR-C5  |  √ | 50e | **45.4**  | 25.1  | 49.3  | 63.8  |
>
>
> According to the results in the above tables, our Group DETR still shows significant improvements over the baselines with longer training times. We will add the results to Table 1 and Table 2 when we finish all the experiments with the longer training settings.
>
> ___

---

> > ### Author Response · Authors · 2022-11-17
> > **Response to reviewer ySfN (2/2)**
> >
> > **Q4: It is not clear why adding new groups improve the performance since there are more negative queries forcing in-object queries to be negatives as well. The author should have a better explanation.**
> >
> > **A4:** It is true that the numbers of both positive queries and negative queries increase when adding new groups.
> >
> > We would like to clarify the meaning of "positive" query in the DETR framework. A query is selected as positive if it wins in terms of the matching cost: it is a better query to match with one ground-truth object than other queries. In other words, a positive query is a winner query, and it is possible that other queries may have good box predictions, and the so-called “classification” score (essentially a mixture of the score belonging to one category and the score that it matches with the ground-truth compared to other queries) is low because it is a loser when competing with other queries that also have good box predictions. In summary, a positive query is also about the comparison to other queries, and the DETR decoder also learns the comparison between queries.
> >
> > Adding new groups introduces more winners for one ground-truth, and thus the DETR decoder is more capable of learning the "comparison", resulting in improved performances.
> >
> > ___
> >
> > **Q5: Failure cases and limitations of the proposed methods.**
> >
> > **A5:** Thanks for the suggestions. We admit that our Group DETR is not perfect and have listed some limitations in the paper (e.g., the increments of training memory and training time), which may need further improvement and engineering optimization. And we will add failure cases in the paper.

---

> > ### Comment · Reviewer_ySfN · 2022-11-18
> > **Response to authors**
> >
> > Thanks the authors for your response. However,
> > * For the A2, I still see that Group-DETR is another option for improving the performance of the DETR-like detectors a little bit. It is not a clear winner with a significant trade-off between performance and time-memory cost.
> > * For the A4, it has not answered my question yet on how more positive and negative queries improve the convergence. Given a GT box, instead of having 1 positive and 20 negative queries, right now you have K positive and 20*K negative queries. This also gives more confusion, e.g., a query is negative in a group while it can be positive in another group due to no better queries in that group. I think just a simple intuitive explanation is not enough to explain the most important contribution of this paper.  Your answer that more winners give more capable of learning the "comparison" is originally not stated in the paper and does not have any proof or explanation on that.
> >
> > My overall thought is that your paper needs to improve a lot in explaining why your method works not just presenting how it works.

---

> > > ### Author Response · Authors · 2022-11-18
> > > **Response to reviewer ySfN**
> > >
> > > We thank the reviewer for the response and address the concerns below.
> > >
> > > ___
> > >
> > > **Q1: For the A2, I still see that Group-DETR is another option for improving the performance of the DETR-like detectors a little bit. It is not a clear winner with a significant trade-off between performance and time-memory cost.**
> > >
> > > **A1:** Thanks for agreeing that our Group DETR can improve the performance of the DETR-like detectors. And we want to summarize the characteristics of our Group DETR here:
> > >
> > > - **Group DETR aims to alleviate the slow convergence issue in DETR-like detectors from the perspective of assignment methods, which provides a new aspect for this issue.** It brings important value to the community and helps researchers better understand DETR-like detectors.
> > >
> > > - **Group DETR is a simple and effective method for various DETR-like detectors.** According to the results in the paper and in the responses above, Group DETR brings significant improvements to Conditional DETR, DAB-DETR, DN-DETR, and DAB-Deformable DETR compared with the baseline with the same epochs or with the same training time. When applying to the highly-optimized DINO, Group DETR can also bring non-negligible gains, even with a strong backbone, Swin-Large. Besides, Group DETR also works on Multi-view 3D Object Detection and Instance Segmentation. All these results show the effectiveness of Group DETR.
> > >
> > > - **Group DETR brings no extra cost in model inference.** Group DETR only adds more groups during the training phase and shares the exact same architecture and process as the original model in inference. We agree with the reviewer that Group DETR is not perfect and increase the training memory. But this limitation can be alleviated with better implementation and engineering optimization to make it affordable, e.g., applying FlashAttention[1].
> > >
> > > [1] Dao T, Fu D Y, Ermon S, et al. FlashAttention: Fast and Memory-Efficient Exact Attention with IO-Awareness[J]. arXiv preprint arXiv:2205.14135, 2022.
> > >
> > > Overall, our method is not perfect, but we believe it still makes some contribution to the community and hope the above responses will facilitate the reviewers' assessment of this paper.
> > >
> > > ___
> > >
> > >
> > > **Q2: For the A4, it has not answered my question yet on how more positive and negative queries improve the convergence. Given a GT box, instead of having 1 positive and 20 negative queries, right now you have K positive and 20*K negative queries. This also gives more confusion, e.g., a query is negative in a group while it can be positive in another group due to no better queries in that group. I think just a simple intuitive explanation is not enough to explain the most important contribution of this paper. Your answer that more winners give more capable of learning the "comparison" is originally not stated in the paper and does not have any proof or explanation on that.**
> > >
> > >
> > > **A2:** Let’s give more explanation about comparison/winner and eliminate “confusion”. The “winner” perspective is related to the machine learning problem, “order learning”. In the pairwise setting for order learning, one feasible method (e.g., [2]) categorizes the relationship between two instances, where one sample could be greater than another sample in one pair and smaller than the third sample in another pair. The “greater” and “smaller” relationship in different pairs for one sample is similar to  “a query is negative in a group while it can be positive in another group due to no better queries in that group”. Accordingly, there is no confusion in our Group DETR.
> > >
> > > In addition, we would like to provide more about the “winner” process in DETR. In DETR, the NMS process is not necessary. It is known that NMS is actually a comparison process, taking the winner and leaving the loser out. DETR essentially models the comparison process and the winner-taking process through decoder self-attention (which collects other queries), and bipartite matching (that finds a winner as the training target). Hope this help answer your question.
> > >
> > > [2] Kyungsun Lim, Nyeong-Ho Shin, Young-Yoon Lee, Chang-Su Kim: Order Learning and Its Application to Age Estimation. ICLR 2020
> > >
> > > Moreover, we want to make clarification on our contributions. The most important contributions of our Group DETR are:
> > >
> > > - We investigate the slow convergence issue in DETR-like detectors from the perspective of assignment methods.
> > >
> > > - We propose a group-wise one-to-many assignment to alleviate the slow convergence issue and improve the performances of various DETR-like detectors.
> > >
> > > The above explanations help us better understand our method. We will add these explanations to the paper.

---

> ### Author Response · Authors · 2022-12-06
> **Looking forward to post-rebuttal discussions**
>
> Dear Reviewer ySfN,
>
> We sincerely appreciate your time and effort in reviewing our paper, which will help us to improve our final paper!
>
> Since the deadline for discussion is approaching, we are pleased to provide additional clarification you may need. In our last response, we have carefully considered your comments and provided a detailed response, which is summarized below.
>
> - We highlighted the merits of our Group DETR and our main contributions to the community.
> - We provided more explanations that the comparison/winner perspective for the DETR decoder is related to “order learning”.
>
> We hope that the above explanations have addressed your concerns. Please let us know if there is any further clarification we can provide as well. Thank you for your time!
>
> Best,
>
> Authors

---

### Official Review · Reviewer_5yMc · 2022-10-23

**Confidence:** 5
**Correctness:** 3
**Technical Novelty And Significance:** 2
**Empirical Novelty And Significance:** 2
**Recommendation:** 5

**Clarity, Quality, Novelty And Reproducibility:**

## Clarity
The paper is clear to follow.

## Quality
The experimental results is complete and sufficient, and the quality is thus not bad.

## Novelty
The proposed method can be seen as another version of Denoising DETR, which somehow limits its novelty.
Furthermore, the paper does not provide a convincing explanation about why it works.

## Reproducibility
Some details about GroupDETR + DINO is missing, which makes some results cannot be easily reproduced.

**Strength And Weaknesses:**

## Strength
1. The paper is well-written and easy to follow
2. The idea is simple yet effective, which only use multiple group of queries to train the decoder.
3. The experimental results show that the idea can be applied to and improve many kinds of detection transformers.

## Weakness
1. The explanation about why this method could work is not sufficient, i.e., the reason why the authors do this modification and why it works need further explanation. For now, we only know that it could work, but providing more about why could bring more insights and values to the community.
2. The explanation of the method increases the number and diversity of positive samples seem can only to explain it can help the model coverage faster. But the paper shows that it still works when training 50 epochs. Will the method still work for a longer training schedule (, e.g., by 150 or 300 epochs)? If yes, does the explanation claimed in the paper still make sense?
3. How to apply the method to some dense version DETRs (such as DINO) is not described in detail in the paper. The paper only provides experiment results for the combination of DINO and Group DETR. Furthermore, given the minor improvements over DINO,  can the explanation that increasing the number and diversity of positive samples still holds for these dense version DETRs?
4. There might be some contradiction in the explanation in the last several sentences in Sec 4.4. For example, although all similar queries in different groups are positive samples in Group DETR, the positive query in one group may also be negative samples in another group (or in other groups, there might be some queries that are very similar to the positive query in the current group), which is not shown in Figure 6.
5. Typos: group-truth matching -> ground-truth matching in the last sentence of the 2nd paragraph in Sec.1.

**Summary Of The Paper:**

This paper proposes a simple and effective method to accelerate the convergence speed and improve the performance of detection transformers, which only need to randomly initialize different groups of queries and use them to train the decoder separately. The method can be applied to all existing detection transformers and exhibits consistent improvements.

**Summary Of The Review:**

Overall, this paper adopts a simple yet effective strategy to train detection transformers and brings consistent improvements in both performance and convergence speed on multiple kinds of detection transformers.
However, the paper does not give a sufficient explanation or insights about how and why it works, which limits its contribution and makes the paper more like a technical report rather than a paper for this venue.
Furthermore, some implementation details are missing.

---

> ### Author Response · Authors · 2022-11-17
> **Response to reviewer 5yMc (1/2)**
>
> Thanks a lot for your time and feedback. We give the responses to the concerns below.
>
> ___
>
> **Q1: The explanation about why this method could work is not sufficient, i.e., the reason why the authors do this modification and why it works need further explanation. For now, we only know that it could work, but providing more about why could bring more insights and values to the community.**
>
> **A1:** We agree with the Reviewer that explaining more about our motivation and understanding will provide insights to the community. We provide detailed explanations below.
>
> - In the one-to-one assignment, one ground-truth object is only assigned to one positive object query. While the one-to-many assignment can produce more positive object queries for each object, which we think may bring benefits to DETR training.
> - To verify the hypothesis, we conduct experiments with Conditional DETR and quickly achieve improved results with the one-to-many assignment in 12 epochs and 50 epochs of training. The evidence suggests that the one-to-many assignment can help with model training convergence. However, directly applying the one-to-many assignment to DETR requires NMS, which breaks down the important end-to-end training properties of DETR.
> - To explore multiple positive queries for each object and avoid duplicate prediction, we split the assigned positives into multiple independent groups and keep the one-to-one assignment in each group. We call it a group-wise one-to-many assignment. The results show that we can achieve satisfactory results and achieve end-to-end with group-wise one-to-many assignments. Note that we find the result of our group-wise one-to-many assignment without NMS is very similar to the one-to-many assignment with NMS when adopting a 50-epoch training schedule (as shown in Table 6).
>
> According to the above analysis and the results in the paper, we propose that our Group DETR can speed up the training convergence of DETR-based detectors.
>
> ___
>
> **Q2: The explanation of the method increases the number and diversity of positive samples seem can only to explain it can help the model converge faster. But the paper shows that it still works when training 50 epochs. Will the method still work for a longer training schedule (e.g., 150 or 300 epochs)? If yes, does the explanation claimed in the paper still make sense?**
>
> **A2:** Thanks for agreeing with our explanation that Group DETR can help the model converge faster. In this paper, we follow the usage of "converge faster" in Deformable DETR and Conditional DETR, which means achieving comparable results with fewer training epochs than the baseline model. But it does not represent that longer training will not improve performance.
>
> According to previous methods (Deformable DETR and Conditional DETR), longer training schedules (150 epochs or 108 epochs) give higher detection results. It is reasonable that Group DETR works when training 50 epochs.
>
> | Method  | Epochs | w/ Group DETR  | mAP  | AP$_s$ | AP$_m$ | AP$_l$  |
> | ------------ | ------------ | ------------ | ------------ | ------------ | ------------ | ------------ |
> | Conditional DETR-C5  |  108 | $\times$ | 43.0  | 22.7  | 46.7  | 61.5  |
> | Conditional DETR-C5  |  108 |  √ | **44.0(+1.0)**  | 24.0  | 47.9  | 61.7  |
>
> Moreover, we follow the reviewer's advice to conduct experiments with a longer schedule (108 epochs) on Conditional DETR-C5. Group DETR can also give a 1.0 mAP gain to Conditional DETR-C5 in this setting. The results again validate that Group DETR helps the model converge faster.
>
> ___
>
> **Q3: How to apply the method to DINO is not described in detail in the paper.**
>
> **A3:** As we state in the paper, every group in Group DETR adopts the same process and implementation. The only difference among different groups is the value of used object queries.
>
> For the baseline DINO, which adopts a two-stage process to generate proposals and object queries and apply contrastive denoising queries, we also add these parts in each group. In detail, we construct 11 independent pairs of classification and regression heads to generate 11 groups of proposals and object queries. And we also add independent contrastive denoising queries in each group. Thus, the process and implementation keep the same for each group. We will release the code and trained models for reproduction.
>
> ___

---

> > ### Author Response · Authors · 2022-11-17
> > **Response to reviewer 5yMc (2/2)**
> >
> > **Q4: Given the minor improvements over DINO, can the explanation that increasing the number and diversity of positive samples still holds for DINO?**
> >
> > **A4:** DINO is a highly-optimized DETR variant. It adopts a two-stage process with mixed query selection, adds contrastive denoising queries (CDN), and proposes a look-forward-twice (LFT) scheme based on DAB-Deformable-DETR. It already speeds up the DETR training convergence and achieves 49.4 mAP with ResNet-50 as the backbone in 12 epochs.
> >
> > It is nontrivial to achieve gains on it with respect to speeding up training convergence. Group DETR brings a 0.7 mAP gain to DINO with 12 epochs and a 0.4 mAP gain with 36 epochs. These gains are non-negligible. Moreover, when using a strong backbone like Swin-Large, Group DETR also brings a 0.4 mAP gain to DINO. The results and analysis show the effectiveness of Group DETR.
> >
> >
> > ___
> >
> > **Q5: There might be some contradiction in the explanation in the last several sentences in Sec 4.4. For example, although all similar queries in different groups are positive samples in Group DETR, the positive query in one group may also be a negative sample in another group, which is not shown in Figure 6.**
> >
> > **A5:** This is a good point. It is possible that one positive query in our group may be a negative sample in another group.
> >
> > We would like to clarify the meaning of "positive" query in the DETR framework. A query is selected as positive if it wins in terms of the matching cost: it is a better query to match with one ground-truth object than other queries. In other words, a positive query is a winner query, and it is possible that other queries may have good box predictions, and the so-called “classification” score (essentially a mixture of the score belonging to one category and the score that it matches with the ground-truth compared to other queries) is low because it is a loser when competing with other queries that also have good box predictions. In summary, a positive query is also about the comparison to other queries, and the DETR decoder also learns the comparison between queries.
> >
> > From one-to-one assignment to our group-wise one-to-one assignment, our approach introduces more winners for one ground-truth, and thus the DETR decoder is more capable of learning the "comparison". As a result, there is no contradiction.
> >
> > ___
> >
> > **Q6: Typos: group-truth matching should be ground-truth matching in Sec 1.**
> >
> > **A6:** Thanks for pointing it out. We will fix it.
> >
> > ___

---

> ### Author Response · Authors · 2022-12-06
> **Looking forward to post-rebuttal discussions**
>
> Dear Reviewer 5yMc,
>
> We sincerely appreciate your time and effort in reviewing our paper, which will help us to improve our final paper!
>
> Since the deadline for discussion is approaching, we are pleased to provide additional clarification you may need. In our previous response, we have carefully considered your comments and provided a detailed response, which is summarized below.
>
> - We provided detailed explanations about our motivation and insights in Group DETR.
> - We made explanations about the reason that Group DETR still works with 50 epochs and added a comparison with a longer training schedule.
> - We detailed the implementations of Group DETR on DINO.
> - We made clarifications for the positive queries and provided detailed explanations about the learning of the DETR decoder.
>
> We hope that the above experiments and explanations have addressed your concerns. Please let us know if there is any further clarification we can provide as well. Thank you for your time!
>
> Best,
>
> Authors

---

### Official Review · Reviewer_MZFA · 2022-10-24

**Confidence:** 5
**Clarity, Quality, Novelty And Reproducibility:** The quality, clarity and originality …
**Correctness:** 1
**Technical Novelty And Significance:** 2
**Empirical Novelty And Significance:** 3
**Recommendation:** 5

**Strength And Weaknesses:**

Strengths
++ The proposed training method is conceptually simple and can be adopted by different DETR-like methods.
++ There are adequate experiments, including 2D and 3D object detection, instance segmentation. The proposed method can bring benefits to different extend.
++ The paper is easy to follow.

Weakness
-- The technical novelty is limited. Similar to DN-DETR, the motivation of group DETR is simply adding more training to the decoder. Moreover, the choice of retaining the first group seems very hand-designed and lack a theoretical explanation.
-- There are some key experiments missing. First, for a baseline (a standard DN-DETR, for example), what is the performance using K*N queries, comparing to group DETR using K groups of N query/group?  Second, although the total training epochs are the same between group DETR and its counterparts, the training time increases for group DETR (~15% increase). So, what is the performance of training a baseline with 15% more training epochs? In this case, the total training time is comparable, rather than the training epochs.
-- The increase in GPU memory (i.e. almost double) is significant.

**Summary Of The Paper:**

This paper proposes a training approach to improve the performance of DETR-like object detectors. Namely, it add more query groups as the decoder inputs, and each group of queries is completely independent of each other during training. It means that the original one-to-one label assignment is adopted for each group. At inference, only the first group of queries is retained and the rest are discarded. The paper conducts experiments with a few DETR-like detectors and show various improvements over its original counterpart.

**Summary Of The Review:**

As stated in the strengths and weakness, this paper shows some improvements over its counterparts, however, the proposed group queries seems more like an engineering trick rather than a novelty. Furthermore, it lacks some important experiments.

---

> ### Author Response · Authors · 2022-11-17
> **Response to reviewer MZFA (1/2)**
>
> Thanks a lot for your time and feedback. We give the responses to the concerns below.
>
> ___
>
> **Q1: The technical novelty is limited. Similar to DN-DETR, the motivation of Group DETR is simply adding more training to the decoder.**
>
> **A1:** We make a clarification on the differences between Group DETR and DN-DETR:
>
> - The motivation of Group DETR and DN-DETR is different. Group DETR aims to explore assignment methods to speed up DETR-based methods training convergence and introduce the group-wise one-to-many assignment. While DN-DETR focuses on stabilizing the prediction and ground-truth matching by proposing an auxiliary query denoising task.
> - The implementation of Group DETR and DN-DETR is different. Group DETR constructs multiple groups of randomly initialized object queries. The object queries in each group are independent of the ones in other groups and can be used to perform model inference. DN-DETR performs query denoising on the noised ground-truth boxes with designed noises and can not be used for model inference.
> - Combining Group DETR with DN-DETR can further improve the performance of DN-DETR, implying that these two approaches are complementary (as discussed in Table 7).
>
> ---
>
> **Q2: The choice of retaining the first group seems very hand-designed and lacks a theoretical explanation.**
>
> **A2:** In Group DETR, all groups are equally important and achieve similar results (the std of 11 groups is 0.07 according to the results in Table 5, which is relatively small). When we talk about the first group, we identify the index of the group in our implementation. Actually, any single group is ok and will not affect the result.
>
> ---
>
> **Q3: For a baseline (a standard DN-DETR, for example), what is the performance using K * N queries, comparing to group DETR using K groups of N query/group?**
>
> **A3:** For K * N queries, there are two possible implementations:
>
> - **Implementation A** is increasing the object queries from N to K * N and using K * N object queries in both training and inference. We conduct an experiment with this setting based on DN-DETR-C5 and train the model for 12 epochs. The results and comparisons are provided below. Our Group DETR outperforms the model with more object queries. Note that using K * N object queries in the baseline model increases the computational costs in both training and inference. While Group DETR brings no extra cost to the model in model inference.
>
>
> | Implementations | Method  | **Queries for Inference** | w/ Group DETR  | mAP  | AP$_s$ | AP$_m$ | AP$_l$  |
> | ------------ | ------------ | ------------ | ------------ | ------------ | ------------ | ------------ | ------------ |
> | Baseline  | DN-DETR-C5  |  300 | $\times$ | 38.6  | 17.9  | 41.6  | 57.7  |
> | **Implementation A**  | DN-DETR-C5  |  3300 | $\times$ | 39.1  | 19.5  | 42.4  | 58.0  |
> | Our Group DETR  | DN-DETR-C5  |  300 | √ | **40.6**  | 19.8  | 43.9  | 59.4  |
>
> - **Implementation B** is increasing the number of denoising queries to K * N in DN-DETR. For this setting, we have provided the results of DN-DETR with different numbers of denoising queries in Appendix B Table 8. The results suggest that increasing the number of denoising queries can not achieve significant improvements and show unstable performances. The table below gives the results of 3300 denoising queries, which is inferior to our Group DETR (40.6 mAP).
>
>
> | Implementations | Method  | **Denoising Queries** | w/ Group DETR  | mAP  | AP$_s$ | AP$_m$ | AP$_l$  |
> | ------------ | ------------ | ------------ | ------------ | ------------ | ------------ | ------------ | ------------ |
> | Baseline  | DN-DETR-C5  |  100 | $\times$ | 38.6  | 17.9  | 41.6  | 57.7  |
> | **Implementation B**  | DN-DETR-C5  |  3300 | $\times$ | 38.7  | 18.9  | 42.0  | 57.3  |
> | Our Group DETR  | DN-DETR-C5  |  100 | √ | **40.6**  | 19.8  | 43.9  | 59.4  |
>
> ---

---

> > ### Author Response · Authors · 2022-11-17
> > **Response to reviewer MZFA (2/2)**
> >
> > **Q4: Although the total training epochs are the same between group DETR and its counterparts, the training time increases for group DETR (~15\% increase). So, what is the performance of training a baseline with 15\% more training epochs? In this case, the total training time is comparable, rather than the training epochs.**
> >
> > **A4:** Thanks for the constructive suggestion. We conduct experiments on Conditional DETR, DAB-DETR, and DN-DETR. We train these baselines for longer training times, 15 epochs and 60 epochs, which give 25\% and 20\% more training epochs than 12 epochs and 50 epochs, respectively. The results are shown in the tables below.
> >
> > | Method  | w/ Group DETR | epochs  | mAP  | AP$_s$ | AP$_m$ | AP$_l$  |
> > | ------------ | ------------ | ------------ | ------------ | ------------ | ------------ | ------------ |
> > | Conditional DETR-C5  |  $\times$ | 12e | 32.6  | 14.7  | 35.0  | 48.3  |
> > | Conditional DETR-C5  |  $\times$ | **15e** | 34.4  | 15.1  | 37.3  | 51.3  |
> > | Conditional DETR-C5  |  √ | 12e | **37.6**  | 18.2  | 40.7  | 55.9  |
> > ||
> > | DAB-DETR-C5  |  $\times$ | 12e | 35.2  | 16.7  | 38.6  | 51.6  |
> > | DAB-DETR-C5  |  $\times$ | **15e** | 36.3  | 17.1  | 39.4  | 52.5  |
> > | DAB-DETR-C5  |  √ | 12e | **39.1**  | 19.7  | 42.5  | 56.8  |
> > ||
> > | DN-DETR-C5  |  $\times$ | 12e | 38.6  | 17.9  | 41.6  | 57.7  |
> > | DN-DETR-C5  |  $\times$ | **15e** | 39.3  | 18.2  | 42.5  | 58.2  |
> > | DN-DETR-C5  |  √ | 12e | **40.6**  | 19.8  | 43.9  | 59.4  |
> >
> >
> > | Method  | w/ Group DETR | epochs  | mAP  | AP$_s$ | AP$_m$ | AP$_l$  |
> > | ------------ | ------------ | ------------ | ------------ | ------------ | ------------ | ------------ |
> > | Conditional DETR-C5  |  $\times$ | 50e | 40.9  | 20.5  | 44.2  | 59.6  |
> > | Conditional DETR-C5  |  $\times$ | **60e** | 41.6  | 21.4  | 45.1  | 60.0  |
> > | Conditional DETR-C5  |  √ | 50e | **43.4**  | 23.0  | 47.3  | 62.3  |
> > ||
> > | DAB-DETR-C5  |  $\times$ | 50e | 42.2  | 21.5  | 45.7  | 60.3  |
> > | DAB-DETR-C5  |  $\times$ | **60e** | 42.9  | 22.8  | 46.4  | 61.9  |
> > | DAB-DETR-C5  |  √ | 50e | **44.5**  | 24.2  | 48.5  | 63.2  |
> > ||
> > | DN-DETR-C5  |  $\times$ | 50e | 44.0  | 23.9  | 47.7  | 62.9  |
> > | DN-DETR-C5  |  $\times$ | **60e** | 44.6  | 24.3  | 48.6  | 63.5  |
> > | DN-DETR-C5  |  √ | 50e | **45.4**  | 25.1  | 49.3  | 63.8  |
> >
> >
> > According to the results in the above tables, our Group DETR still shows significant improvements over the baselines with longer training times. We will add the results to Table 1 and Table 2 when we finish all the experiments with the longer training settings.
> >
> >
> > ---
> >
> > **Q5: The increase in GPU memory is significant.**
> >
> > **A5:** We admit that it is one of the limitations of our Group DETR. But this increment in GPU memory is only in the training phase. In model inference, Group DETR brings no extra cost compared with the original model.
> >
> > ---

---

> ### Author Response · Authors · 2022-12-06
> **Looking forward to post-rebuttal discussions**
>
> Dear Reviewer MZFA.
>
> We sincerely appreciate your time and effort in reviewing our paper, which will help us to improve our final paper!
>
> Since the deadline for discussion is approaching, we are pleased to provide additional clarification you may need. In our previous response, we have carefully considered your comments and provided a detailed response, which is summarized below.
>
> - We provided the differences between DN-DETR and our Group DETR.
> - We explained the inference groups.
> - We added new comparisons with the N * K queries baselines and the 15\% more training epoch baselines.
> - We explained the training memory increment, which can be alleviated with better implementation and engineering optimizations.
>
> We hope that the above experiments and explanations have addressed your concerns. Please let us know if there is any further clarification we can provide as well. Thank you for your time!
>
> Best,
>
> Authors

---

### Decision · Program_Chairs · 2023-01-20

**Decision:**

Reject

**Justification For Why Not Higher Score:**

None of the reviewers are supporting the acceptance of the paper due to lacking insights in the proposed approach.

**Justification For Why Not Lower Score:**

N/A

**Metareview: Summary, Strengths And Weaknesses:**

The paper proposes a training method for the DETR object detection framework called Group DETR. DETR uses a set of queries in training. Group DETR uses K sets of queries in training. However, the test time behavior for both methods is the same. The paper shows Group DETR can be used to improve the performance of various DETR methods.

The paper receives three reviews. All of them consider the paper below the bar. While the reviewers are happy with the consistent performance gain, they do not understand why the method works. They feel the insight side of the paper is lacking. The paper was discussed among the reviewers after the rebuttal. However, the reviewers are not convinced that the paper deserves a better score and still leaning toward rejecting the paper. They feel the paper is basically trading training resources for performance gain.

After analyzing the paper, reviews, and rebuttal, the AC is happy with the consistent performance gain achieved and the simplicity of the method. However, the AC also fails to get an insight into the design of the method. As a result, the AC agrees with the reviewers that the paper is slightly below the bar.

**Summary Of Ac-Reviewer Meeting:**

N/A